# Influenza-associated excess mortality in the Philippines, 2006–2015

**Kent Jason Go Cheng**[1]*, **Adovich Sarmiento Rivera**[2], **Hilton Yu Lam**[3], **Allan Rodriguez Ulitin**[3], **Joshua Nealon**[4], **Ruby Dizon**[5], **David Bin-Chia Wu**[6,7]

**1** Social Science Department, Maxwell School of Citizenship and Public Affairs, Syracuse University, Syracuse, NY, United States of America, **2** Feinberg School of Medicine, Northwestern University, Chicago, Illinois, United States of America, **3** Institute of Health Policy and Development Studies, National Institutes of Health, University of the Philippines Manila, Manila, Philippines, **4** Vaccines Epidemiology and Modeling, Sanofi Pasteur, Singapore, Singapore, **5** Medical Affairs, Sanofi Pasteur, Taguig City, Metropolitan Manila, Philippines, **6** Faculty of Pharmacy and Pharmaceutical Sciences, Monash University Malaysia, Selangor, Malaysia, **7** Asian Centre for Evidence Synthesis in Population, Implementation and Clinical Outcomes, Health and Well-Being Cluster, Global Asia in the 21st Century Platform, Monash University Malaysia, Bandar Sunway, Selangor, Malaysia

* kgcheng@syr.edu

**Data Availability Statement:** Flu-related variables can be freely downloaded from the FluNet website. However, Philippine Data Privacy Act restricts our ability to share potentially identifiable health data so we cannot share the dataset with counts of deaths

## Abstract

Influenza-associated mortality has not been quantified in the Philippines. Here, we constructed multiple negative binomial regression models to estimate the overall and age-specific excess mortality rates (EMRs) associated with influenza in the Philippines from 2006 to 2015. The regression analyses used all-cause mortality as the dependent variable and meteorological controls, time, influenza A and B positivity rates (lagged for up to two time periods), and annual and semiannual cyclical seasonality controls as independent variables. The regression models closely matched observed all-cause mortality. Influenza was estimated to account for a mean of 5,347 excess deaths per year (1.1% of annual all-cause deaths) in the Philippines, most of which (67.1%) occurred in adults aged ≥60 years. Influenza A accounted for 85.7% of all estimated excess influenza deaths. The annual estimated influenza-attributable EMR was 5.09 (95% CI: 2.20–5.09) per 100,000 individuals. The EMR was highest for individuals aged ≥60 years (44.63 [95% CI: 4.51–44.69] per 100,000), second highest for children aged less than 5 years (2.14 [95% CI: 0.44–2.19] per 100,000), and lowest for individuals aged 10 to 19 years (0.48 [95% CI: 0.10–0.50] per 100,000). Estimated numbers of excess influenza-associated deaths were considerably higher than the numbers of influenza deaths registered nationally. Our results suggest that influenza causes considerable mortality in the Philippines–to an extent far greater than observed from national statistics–especially among older adults and young children.

## Introduction

Influenza is a serious public health concern that causes 3–5 million cases of severe illness and about 290,000 to 650,000 deaths worldwide each year [1, 2]. Estimates of the influenza burden in individual countries are needed to formulate public health policies and strategies to control

per day. Data on deaths per day can be obtained from the Vital Statistics Division (VSD) of the Philippine Statistics Authority (PSA). Contact Aurora Reolalas, Chief of VSD at PSA (+632 8461 0500 local 820). On the other hand, meteorological data can be secured from the Climatology and Agrometeorology Division (CAD) of the Philippine Atmospheric, Geophysical and Astronomical Services Administration (PAG-ASA). Contact Edna C. Juanilo, Weather Services Chief of CAD PAG-ASA (ejuanillo@pagasa.dost.gov.ph; +63 02 434 9024/ +63 02 435 1675).

**Funding:** Research funding was provided by Sanofi Pasteur. The funder also provided support in the form of salaries for authors (JN and RD), but did not have any additional role in the study design, data collection and analysis, decision to publish, or preparation of the manuscript. The specific roles of these authors are articulated in the 'author contributions' section. All authors had full access to the data in the study and take responsibility for the integrity of the data and accuracy of its analysis.

**Competing interests:** KJGC, ARU, ASR, HYL, DBCW received fee compensation through grant provided by Sanofi Pasteur for this study. JN holds stock in Sanofi Pasteur. This does not alter our adherence to PLOS ONE policies on sharing data and materials.

influenza. This is particularly important to protect those at greater risk of developing life-threatening influenza complications, such as young children, older adults, and people with chronic illnesses [2, 3]. However, numbers of influenza-attributable deaths are difficult to measure directly because influenza infections are not typically laboratory-confirmed and may not be diagnosed [4]. In addition, direct mortality measurements can miss deaths from secondary complications triggered by influenza infection (e.g., influenza-triggered exacerbation of pre-existing chronic illnesses).

Although the burden of influenza has been extensively evaluated in temperate regions of Europe and North America, it is less well characterized for many countries in Asia. The Philippines is located in Southeast Asia's tropical climate region, which is generally considered an important source of new viruses and global influenza epidemics because of the large and highly interacting human and animal populations [5]. Influenza A and B viruses circulate throughout the year in the Philippines, and there are often multiple annual peaks in influenza activity [6–8]. Circulating influenza strains tend to match the Southern rather than the Northern Hemisphere vaccine strains, hence the Southern Hemisphere influenza vaccine is used in each year's national vaccination program [8]. The burden of influenza in the Philippines is largely unknown. The mean annual influenza incidence rate has been estimated as 5.4 per 1,000 individuals in an urban region of the country, with particularly high incidence (22.6 per 1,000) in young children [6]. However, the rate of influenza-associated mortality has not been quantified. Here, we used negative binomial regression models to estimate the influenza-attributable excess mortality in the Philippines from 2006 to 2015, and compared these influenza mortality estimates with death registry data to quantify under-reporting of influenza deaths.

## Methods

### Study design

This was a retrospective analysis of influenza-associated deaths over the period Jan 1, 2006 to December 31, 2015 in the Philippines. The objective was to estimate the overall and age-specific excess mortality rates (EMRs) associated with influenza. Ethical approval was not required for this analysis of aggregated administrative data.

### Data sources

Weekly all-cause deaths were obtained from the death registration dataset of the Philippine Statistics Authority [9] (data summarized in **S1 Table**). In the Philippines, deaths are certified using International Classification of Diseases Tenth Revision (ICD-10) codes [10]. However, official reporting systems rarely capture every death that occurred; for instance, one study found that in one Philippine province, only 77% of deaths were captured in government records [11]. But since the extent of this under-reporting problem has yet to be examined nationally, we made no adjustment for under-registration of deaths for this study.

Weekly percentages of laboratory-confirmed influenza A and B cases in the Philippines were obtained from the WHO's Global Influenza Surveillance and Response System (GISRS) FluNet database [12] (**S1 Table and S1 Fig**). The GISRS data for the Philippines was collected through passive surveillance of influenza-like illness (ILI) and severe acute respiratory infection cases at sentinel sites located throughout the country. In the surveillance, ILI was defined as an acute respiratory infection with measured fever of ≥38C˚ and cough with onset within the last 10 days while severe acute respiratory infection (SARI) was defined as an acute respiratory infection with history of fever or measured fever of ≥38C˚ and cough with onset within the last 10 days and requires hospitalization [13]. ILI sentinel surveillance sites are health centers and hospital outpatient departments while SARI sentinel surveillance sites are hospital

inpatient departments [8]. Laboratory confirmation of influenza virus from clinical samples was performed at the Research Institute for Tropical Medicine, Metropolitan Manila, Philippines (the WHO-designated National Influenza Center) by real-time reverse-transcription polymerase chain reaction. More details about the surveillance can be found in a previous study [8].

To account for climatic variation in influenza transmission and seasonality, meteorological data (rainfall, mean temperature, and relative humidity) were obtained from the 52 weather stations of the Philippine Atmospheric, Geophysical and Astronomical Services Administration [14]. Fifteen weather stations were excluded because they had more than 6 months of missing data. Weekly nationwide values for rainfall and relative humidity were obtained by averaging data from the 37 included weather stations. The average weekly temperature was calculated by taking the mean of the weekly maximum and weekly minimum, following the logic of the World Meteorological Organization's recommended method of computing for average daily temperature [15].

## Data analyses

Excess mortality associated with influenza was estimated for the overall population and for five age groups (0 to 4 y, 5 to 9 y, 10 to 19 y, 20 to 59 y, and ≥60 y) using negative binomial regression models (regression equations for each age group can be found in the, **S1 Text**). Negative binomial regression was used instead of the Poisson regression since deaths were over-dispersed. Because our study included the time when the Philippines was affected by typhoon Haiyan, a tropical storm resulting in >6,000 deaths, this event was formally included in the models which were based on the following equation:

$$
\begin{aligned}
E[Y_t] = \exp\{ & \beta_0 + \beta_1 t + \beta_2 t^2 + \beta_3 t^3 + \beta_4 t^4 + \beta_5 t^5 + \beta_6 t^6 + \beta_7 [\text{Influenza A}]_t \\
& + \beta_8 [\text{Influenza B}]_t + \beta_9 [\text{Rainfall}]_t + \beta_{10} [\text{Mean Temperature}]_t \\
& + \beta_{11} [\text{Relative Humidity}]_t + \beta_{12} [\text{Haiyan}]_t + \beta_{13} [\text{Pandemic}]_t \\
& + \beta_{14} [\sin(2\pi t/52)] + \beta_{15} [\cos(2\pi t/52)] + \beta_{16} [\sin(2\pi t/26)] \\
& + \beta_{17} [\cos(2\pi t/26)] + e_t \}
\end{aligned}
$$

where t denotes time, $E[Y_t]$ is the expected value of weekly number of all-cause deaths Y, and β values are the coefficients. $\beta_0$ is the intercept; $\beta_1$ to $\beta_6$ account for the polynomial time trends; $\beta_7$ and $\beta_8$ are coefficients associated with the percentage of samples confirmed positive for influenza A and influenza B, respectively; $\beta_9$ to $\beta_{11}$ are coefficients for the meteorological data, rainfall ($\beta_9$), mean temperature ($\beta_{10}$), and relative humidity ($\beta_{11}$); $\beta_{12}$ and $\beta_{13}$ pertain to the coefficients of the dichotomous variable for the Typhoon Haiyan week (week 45 of 2013) and the 2009 flu pandemic, respectively; $\beta_{14}$ and $\beta_{15}$ pertain to annual cyclical terms; $\beta_{16}$ and $\beta_{17}$ pertain to semiannual cyclical terms; and lastly, e is the error term that follows $\exp(e_t) \sim$ Gamma($1/\alpha, \alpha$), and $\alpha$ is the overdispersion parameter. Annual and semiannual cyclical terms were used as seasonality controls since there are two seasons in the Philippines and influenza seasonality is known to be semi-annual, peaking from around June to November [8, 16]. The polynomial time trend and seasonality time trends were entered consecutively to determine the regression equation that best fitted the data. We also included various lags (no lag, one-week lag, and two-week lags) of the flu positivity rates to account for the possible delayed effect of flu on mortality. This iterative process (**S2 Table**) resulted in 84 regressions calculated for each age group. The regression that had the lowest Akaike Information Criterion [17], i.e. the model that provides the best fit was selected (**S3 Table**). Because the number of tested samples was not available for week 16 of 2008, the value was imputed by taking the mean of the data from the week before and the week after. To estimate the excess mortality associated with

influenza, we first calculated the annual predicted all-cause deaths for each age group using the chosen regression model, and then subtracted the annual predicted deaths without influenza A or without influenza B (coefficients for influenza A or B set to zero), as detailed elsewhere [1, 4, 18–20]. The resulting annual average mortality due to influenza A or B were divided by the 2015 total population and multiplied by 100,000 to get the EMR per 100,000 persons. The 95% confidence intervals (CI) for the EMRs were estimated through bootstrapping of residuals and re-estimating the excess mortality (1,500 iterations). The estimated number of influenza deaths was compared to the number of influenza deaths registered by the Philippine Statistics Authority with ICD-10 codes J10 ('influenza due to other identified influenza virus') and J11 ('influenza due to unidentified influenza virus with other respiratory manifestations'). Sensitivity analyses were also performed to investigate the robustness of the estimates by running the same regressions with imputed values for all-cause deaths for week 45 of 2013 (coinciding with Typhoon Haiyan) and for the influenza positivity rate for weeks with ≤1 sample tested or with ≤10 samples tested. Imputed values were the average of the data from the week before and the week after the data point where possible; otherwise, they were the average of the two weeks nearest to the data point to be imputed. All analyses were performed using R version 3.5.1 [21].

## Results

### Descriptive analyses

An average of 485,412 all-cause deaths were registered per year in the Philippines between 2006 and 2015 (Table 1). Around 55% of all-cause deaths were of individuals aged ≥60 years (S1 Table). A mean of 8,418 samples per year were tested for influenza during the same period, and a mean of 1,453 (17.3%) were positive for influenza virus (Table 1). Influenza A accounted for most (78.2%) of the confirmed influenza cases, although influenza B was detected more frequently than influenza A in 2008 and 2013. The proportion of samples testing positive for influenza A or B varied by year. One-third of samples (33.3%) were positive for influenza in the A/H1N1 2009 pandemic season, almost all of which were confirmed as influenza A. Excluding 2009, the proportion of samples positive for influenza A or B varied between 6.9% and 19.2%.

**Table 1. All-cause deaths and laboratory-confirmed influenza cases recorded in the Philippines, 2006–2015.**

| Year | Number of weeks FluNet data was available | All-cause deaths, N | Samples tested, N | Influenza-positive | | Influenza A | | Influenza B | |
|---|---|---|---|---|---|---|---|---|---|
| | | | | n | % | n | % | n | % |
| 2006 | 52 | 439,772 | 5,955 | 557 | 9.4% | 381 | 6.4% | 176 | 3.0% |
| 2007 | 52 | 440,651 | 6,291 | 536 | 8.5% | 504 | 8.0% | 32 | 0.5% |
| 2008[a] | 52 | 458,793 | 11,676 | 807 | 6.9% | 211 | 1.8% | 596 | 5.1% |
| 2009 | 50 | 460,462 | 23,169 | 7,706 | 33.3% | 7,556 | 32.6% | 150 | 0.6% |
| 2010 | 51 | 476,754 | 10,101 | 1,724 | 17.1% | 875 | 8.7% | 849 | 8.4% |
| 2011 | 52 | 497,166 | 9,689 | 894 | 9.2% | 627 | 6.5% | 267 | 2.8% |
| 2012 | 50 | 491,110 | 7,346 | 650 | 8.8% | 403 | 5.5% | 247 | 3.4% |
| 2013 | 51 | 517,767 | 5,052 | 972 | 19.2% | 388 | 7.7% | 584 | 11.6% |
| 2014 | 51 | 535,336 | 3,765 | 522 | 13.9% | 269 | 7.1% | 253 | 6.7% |
| 2015 | 50 | 536,305 | 1,140 | 163 | 14.3% | 149 | 13.1% | 14 | 1.2% |
| Mean | 51.1 | 485,412 | 8,418 | 1,453 | 17.3% | 1,136 | 13.5% | 317 | 3.8% |

[a] Number of samples tested not available for week 16 of 2008.

### All-cause and influenza-associated mortality using the negative binomial regression method

The all-cause mortality predicted in the negative binomial regression replicated the registered weekly all-cause mortality for each age group (Fig 1). In addition, the models fit the data well as they predicted a rise in the mortality rate from 2006 to 2015 among individuals aged 20 to 59 years and ≥60 years. There was a notable peak in registered deaths across age groups in week 45 of 2013 coinciding with Typhoon Haiyan.

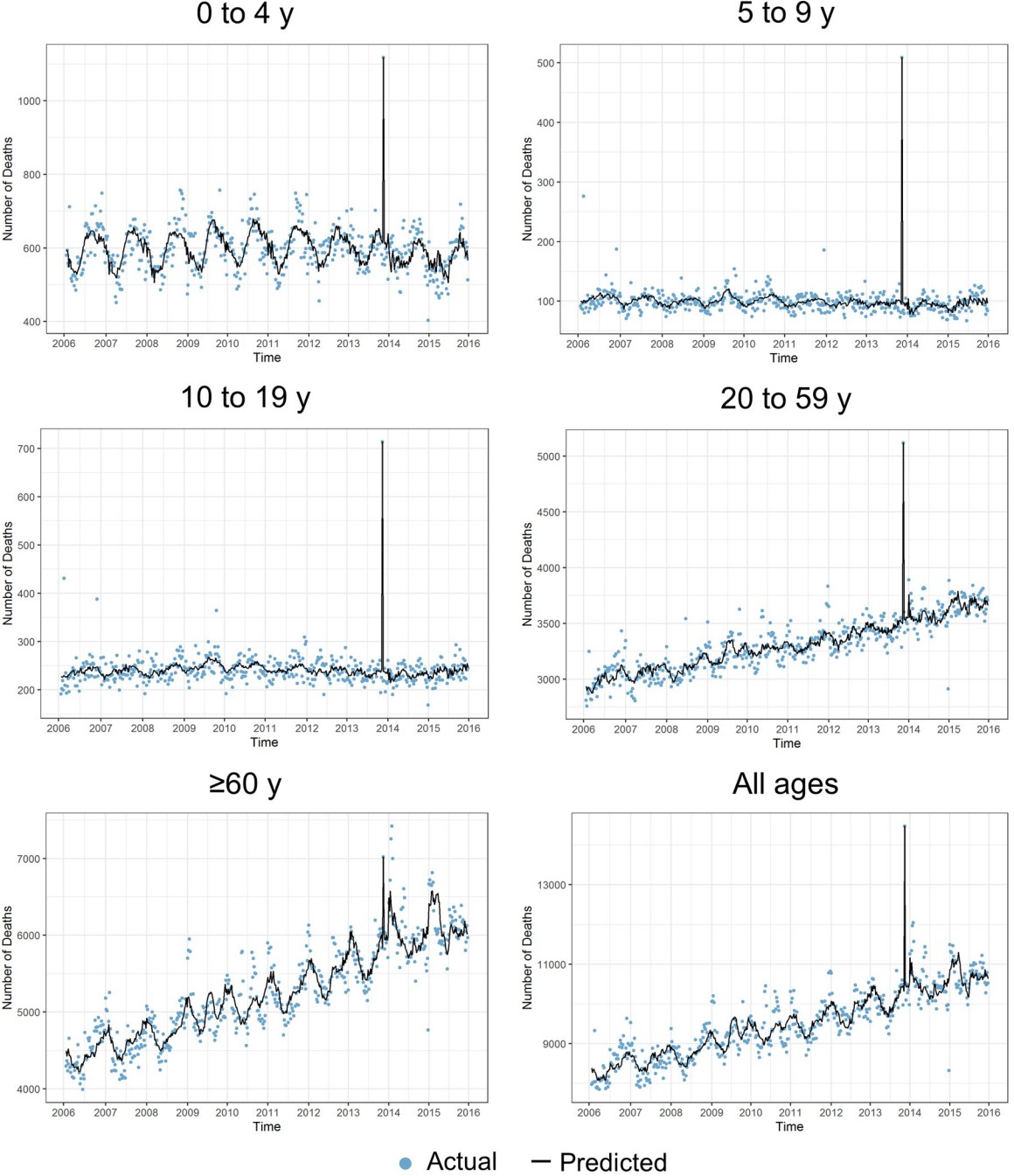

**Fig 1. Actual versus estimated weekly all-cause mortality per age group, 2006–2015.**

Influenza was estimated to account for a mean of 5,347 excess deaths per year (about 1.1% of the average annual all-cause deaths for the study period) in the Philippines over the study period, most of which (67.1%) occurred in adults aged ≥60 years (Table 2). Influenza A accounted for 4,584 (85.7%) of all estimated excess influenza deaths. Although influenza B was estimated to cause fewer influenza deaths overall, it was responsible for about one-third (31.2%) of influenza deaths among 5–9-year-olds and one-fourth (24.1%) of influenza deaths among 0–4 year-olds.

The highest number of influenza-associated deaths was estimated to have occurred in 2009 (n = 8,784), almost all of which were caused by influenza A consistent with the 2009 A/H1N1 pandemic (Table 3). Deaths associated with influenza A or B were higher than average in the years 2013–2015 (n = 7,046–8,666) and were least frequent in 2008 (n = 1,807).

Overall, the annual EMR for influenza A and B-associated deaths was estimated as 5.09 (95% CI: 2.20–5.09) per 100,000 individuals (Table 2). The EMR was highest for individuals aged ≥60 years (44.63 [95% CI: 44.51–44.69] per 100,000), and second highest for children aged <5 years (2.14 [95% CI: 0.44–2.19] per 100,000). The lowest EMR was among individuals aged 10 to 19 years (0.48 [95% CI: 0.10–0.50] per 100,000). The overall annual EMR was 4.37 (95% CI: 1.49–4.37) per 100,000 for influenza A-associated deaths, and 0.73 (95% CI: −2.15–0.73) per 100,000 for influenza B-associated deaths.

A peak in influenza deaths also occurred in the week of Typhoon Haiyan (week 45 of 2013, figure not shown). However, in a sensitivity analysis, the EMRs were not considerably changed by using an imputed value for all-cause deaths at week 45 of 2013 (S4 Table). The EMRs were also similar in other sensitivity analyses using imputed values for weeks with ≤1 sample tested (n = 1 week) and for weeks with ≤10 samples tested (n = 21 weeks).

## Comparison of estimated influenza-associated deaths with national registry data

The estimated annual influenza-associated deaths were compared with influenza deaths registered by the Philippine Statistics Authority (Table 3). For all seasons, and across all age groups, the numbers of estimated excess influenza-associated deaths were considerably greater than those registered in the Philippines. Overall, compared to our estimated mean of 5,347 excess influenza-associated deaths per year, a mean of 96 deaths per year were registered.

## Discussion

Our study provides the first estimate of influenza-associated mortality in the Philippines. The country's influenza-associated EMR was considerable between 2006 and 2015: an estimated

**Table 2. Estimated average annual influenza excess mortality per age group and annual excess mortality rate per 100,000 individuals, 2006–2015.**

| Age group | Influenza A & B | | Influenza A | | Influenza B | |
|---|---|---|---|---|---|---|
| | Mean excess deaths, n (%) | EMR (95% CI) | Mean excess deaths, n (%) | EMR (95% CI) | Mean excess deaths, n (%) | EMR (95% CI) |
| 0 to 4 y | 249 (4.7) | 2.14 (0.44–2.19) | 189 (4.1) | 1.62 (−0.02–1.66) | 60 (7.9) | 0.52 (−1.14–0.55) |
| 5 to 9 y | 169 (3.2) | 1.51 (1.20–1.56) | 116 (2.5) | 1.04 (0.75–1.08) | 53 (6.9) | 0.48 (0.17–0.51) |
| 10 to 19 y | 100 (1.9) | 0.48 (0.10–0.50) | 84 (1.8) | 0.40 (0.04–0.42) | 16 (2.1) | 0.08 (−0.29–0.09) |
| 20 to 59 y | 1,076 (20.1) | 2.02 (2.00–2.03) | 1,049 (22.9) | 1.97 (1.96–1.97) | 27 (3.5) | 0.05 (0.04–0.06) |
| ≥60 y | 3,587 (67.1) | 44.63 (44.51–44.69) | 3,062 (66.8) | 38.15 (38.07–38.18) | 525 (68.8) | 7.08 (6.45–6.57) |
| All ages[a] | 5,347 (−) | 5.09 (2.20–5.09) | 4,584 (−) | 4.37 (1.49–4.37) | 763 (−) | 0.73 (−2.15–0.73) |

Abbreviations: CI, confidence interval; EMR, excess mortality rate per 100,000.

[a] Mean deaths for all ages were derived using the same estimation strategy as the other age groups, and therefore do not equal the sum of all ages.

**Table 3. Estimated excess influenza-associated deaths versus nationally registered influenza deaths per age group.**

| Age group | 2006 | 2007 | 2008 | 2009 | 2010 | 2011 | 2012 | 2013 | 2014 | 2015 | Mean (2006–2015) |
|---|---|---|---|---|---|---|---|---|---|---|---|
| 0 to 4 y | | | | | | | | | | | |
| Estimated | 165 | 176 | 121 | 400 | 277 | 195 | 183 | 367 | 296 | 309 | 249 |
| Registered | 11 | 1 | 4 | 3 | 5 | 1 | 3 | 5 | 0 | 1 | 4 |
| 5 to 9 y | | | | | | | | | | | |
| Estimated | 116 | 109 | 95 | 254 | 195 | 127 | 115 | 281 | 209 | 191 | 169 |
| Registered | 2 | 2 | 3 | 2 | 0 | 1 | 1 | 2 | 0 | 0 | 2 |
| 10 to 19 y | | | | | | | | | | | |
| Estimated | 66 | 74 | 38 | 177 | 99 | 80 | 66 | 129 | 133 | 139 | 100 |
| Registered | 8 | 7 | 5 | 2 | 6 | 3 | 3 | 2 | 2 | 1 | 4 |
| 20 to 59 y | | | | | | | | | | | |
| Estimated | 657 | 824 | 232 | 1,986 | 901 | 866 | 711 | 1,151 | 1,496 | 1,936 | 1,076 |
| Registered | 43 | 36 | 27 | 26 | 21 | 7 | 17 | 29 | 14 | 11 | 24 |
| ≥60 y | | | | | | | | | | | |
| Estimated | 1,979 | 2,317 | 1,192 | 5,740 | 3,291 | 2,823 | 2,492 | 4,822 | 5,233 | 5,978 | 3,587 |
| Registered | 99 | 81 | 85 | 58 | 52 | 26 | 47 | 121 | 32 | 35 | 64 |
| All ages | | | | | | | | | | | |
| Estimated | 3,105 | 3,602 | 1,807 | 8,784 | 4,984 | 4,222 | 3,692 | 7,046 | 7,563 | 8,666 | 5,347 |
| Registered | 163 | 127 | 124 | 91 | 84 | 38 | 71 | 159 | 48 | 48 | 96 |

The estimated number of influenza deaths was compared to influenza deaths registered by the Philippine Statistics Authority with ICD-10 code J11.1 ('influenza due to unidentified influenza virus with other respiratory manifestations').

5.09 influenza-attributable deaths occurred per 100,000 persons each year, and influenza was the cause of approximately one in every 100 deaths. A disproportionate percentage of influenza-attributable excess deaths (67.1%) occurred among individuals aged 60 years or older, considering that this age group represented just 7% of the total population [22].

The age group-specific EMRs estimated in our study are consistent with previous Filipino influenza mortality estimates by Iuliano et al., who estimated mortality rates per 100,000 of 4.0 (95% CI: 0.6–8.5) for those aged less than 65 years and 50.8 (95% CI: 12.9–96.1) for 65–74 year-olds [1]. However, these earlier estimates for the Philippines were extrapolated using EMRs of neighboring countries, and may not fully reflect the local parameters used in our study. On the other hand, our EMR for children aged <5 years of 2.1 per 100,000 is close to a recent meta-analyses by Wang et al. who found that EMR for the said age group for low-middle income countries is 1.7 [23]. The age-specific EMRs in our study are also aligned with those reported in other tropical countries where Southern Hemisphere influenza strains usually dominate [20, 24–26]. For instance, in Thailand, annual influenza-associated EMRs were estimated to be highest for those aged ≥65 years (42 per 100,000) [25] and in Western Kenya, the influenza EMR was estimated highest among those aged 50 years and older (74.0 per 100,000) and second highest in children aged less than 5 years (22.2 per 100,000) [26].

Our negative binomial regression was predictive of all-cause mortality in the Philippines, providing confidence in the modeling approach used. For example, we correctly predicted a considerable increase in influenza A deaths during 2009, coinciding with the A/H1N1pdm09 influenza pandemic that spread in most of Asia [8, 16]. Our regression models also projected a peak in influenza deaths in the week of typhoon Haiyan. Although this peak could be related to struggling health services and increased transmission of infections following the typhoon [27], it might also be partly influenced by the increased all-cause deaths at this time. By nature of regression analyses, trends of both dependent and independent variables are summarized

by a fixed coefficient. Therefore, the peak in all-cause deaths–the dependent variable–could have translated to upward trends in estimated influenza deaths, even if the influenza positivity rates of tested samples were not elevated during the same period. Nonetheless, the overall EMRs were largely unaffected by this exceptional event since we controlled for the Typhoon Haiyan week in the main analyses and in the sensitivity analyses, where we used an imputed value to replace the peak in all-cause deaths for the Haiyan week. A similar effect from increasing all-cause deaths in some age groups may explain the upward trend in influenza deaths from 2013–2015.

For each age group, we estimated significantly more influenza deaths than the numbers registered in national statistics. Although this might partly follow unreliable use of ICD-10 during death registration and low level of diagnostic confirmation [28], we suspect that our greater estimates largely result from the additional deaths from underlying health conditions exacerbated by influenza (e.g., cardiovascular conditions) besides those caused by influenza directly (e.g., influenza that leads to pneumonia). Our findings suggest that influenza mortality in the Philippines is greater than previously thought, and this information may help encourage improvements in the national influenza surveillance and public health programs. Increasing influenza vaccination coverage among risk groups could be an effective way to reduce influenza-attributable mortality in the country [3]. Indeed, vaccination coverage was only 2.3% for adults aged 60 years and older during the Philippines' last public influenza vaccination program [29].

Our regression analyses had several limitations. First, because of missing FluNet data and to minimize selection bias, we were unable to estimate deaths associated with individual influenza A subtypes or influenza B lineages. Second, few ILI cases in the FluNet database had laboratory confirmation of influenza virus, likely because virologic testing is expensive and of limited clinical value. This may have introduced some selection bias that we could not adjust for, and suggests that the influenza positivity rate data is not representative. Nonetheless, in our sensitivity analyses, the EMRs were largely unaffected by imputed influenza positivity rates for weeks with ≤1 or ≤10 specimens tested, implying that our results are robust despite the limited influenza activity data. Third, FluNet data may not be nationally-representative. Sentinel surveillance sites were only present in 13 out of 17 regions in the country [8] and there is no indication that sites were distributed evenly across the country. Despite its limitations, FluNet data is the only source of influenza activity data of its scope in the country to date. Fourth, because of missingness of the data, we had to drop 15 out of 52 weather stations for the meteorological variables. We believe that this is not an issue since the meteorological variables were merely controls, not the independent variable of interest. Finally, unlike other studies [4, 30], our analyses did not control for infections caused by respiratory syncytial virus because this data was not obtainable for the Philippines. Since respiratory syncytial viruses co-circulate with influenza viruses [31], our results might have overestimated influenza mortality, particularly in the younger age groups [32].

## Conclusions

Our results suggest that the numbers of excess deaths attributable to influenza in the Philippines are considerably greater than those recorded in the national death registry, especially among older adults and young children. These findings underscore the importance of prioritizing older adults and children less than 5 years of age for influenza vaccination, in line with recommendations by the World Health Organization [3].

## Supporting information

**S1 Text. Negative binomial regression equations for each age group.**
(DOCX)

**S1 Table. Average weekly all-cause deaths and influenza-positive samples, 2006–2015.**
(DOCX)

**S2 Table. Model fitting algorithm.**
(DOCX)

**S3 Table. Selected negative binomial regression models.**
(DOCX)

**S4 Table. Sensitivity analyses.**
(DOCX)

**S1 Fig. % Influenza A and influenza B to total samples tested.**
(TIF)

**S2 Fig. Meteorological controls.**
(TIF)

**S3 Fig. Actual all-cause mortality and predicted all-cause mortality with influenza set to zero.**
(TIF)

## Acknowledgments

The authors are deeply indebted to the Philippine Statistics Authority and the Philippine Atmospheric, Geophysical and Astronomical Services Administration for providing us with the data needed to run the analyses. The authors would like to thank the research assistance of the following people: Camille Princess Aguila, Jojana Christine General, Jennifer Ildefonzo, Jodie Mae Penado, and Justine Marjorie Tiu. We would also like to acknowledge the valuable feedback from the anonymous reviewers; the remaining errors are the authors' alone. Medical writing assistance was provided by Drs. Jonathan Pitt and Surayya Taranum (4Clinics, France, Paris).

## Author Contributions

**Conceptualization:** Kent Jason Go Cheng, Adovich Sarmiento Rivera, Hilton Yu Lam, Allan Rodriguez Ulitin, Joshua Nealon, Ruby Dizon.

**Data curation:** Kent Jason Go Cheng, Adovich Sarmiento Rivera.

**Formal analysis:** Kent Jason Go Cheng, Adovich Sarmiento Rivera, David Bin-Chia Wu.

**Funding acquisition:** Hilton Yu Lam, Joshua Nealon, Ruby Dizon.

**Investigation:** Kent Jason Go Cheng, Adovich Sarmiento Rivera, Hilton Yu Lam, Joshua Nealon, Ruby Dizon.

**Methodology:** Kent Jason Go Cheng, Adovich Sarmiento Rivera, Joshua Nealon, David Bin-Chia Wu.

**Project administration:** Kent Jason Go Cheng, Adovich Sarmiento Rivera, Allan Rodriguez Ulitin.

**Resources:** Allan Rodriguez Ulitin.

**Software:** Kent Jason Go Cheng, Adovich Sarmiento Rivera.

**Supervision:** Kent Jason Go Cheng, Adovich Sarmiento Rivera, Allan Rodriguez Ulitin, Joshua Nealon, Ruby Dizon.

**Validation:** Kent Jason Go Cheng, Adovich Sarmiento Rivera, Hilton Yu Lam, Allan Rodriguez Ulitin, Joshua Nealon, Ruby Dizon, David Bin-Chia Wu.

**Visualization:** Kent Jason Go Cheng, Adovich Sarmiento Rivera, Hilton Yu Lam, Allan Rodriguez Ulitin, Joshua Nealon, Ruby Dizon, David Bin-Chia Wu.

**Writing – original draft:** Kent Jason Go Cheng.

**Writing – review & editing:** Kent Jason Go Cheng, Adovich Sarmiento Rivera, Hilton Yu Lam, Allan Rodriguez Ulitin, Joshua Nealon, Ruby Dizon, David Bin-Chia Wu.

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
