## [Decision Letter · Decision Letter 0]

16 Oct 2019

PONE-D-19-24478

Influenza-associated excess mortality in the Philippines, 2006-2015

PLOS ONE

Dear Mr. Cheng,

Thank you for submitting your manuscript to PLOS ONE. After careful consideration, we feel that it has merit but does not fully meet PLOS ONE’s publication criteria as it currently stands. Therefore, we invite you to submit a revised version of the manuscript that addresses the points raised during the review process.

Two reviewers agree that your manuscript needs a major revision, so please address all of their comments on methodological and statistical modelling issues before resubmitting.

We would appreciate receiving your revised manuscript by Nov 30 2019 11:59PM. To enhance the reproducibility of your results, we recommend that if applicable you deposit your laboratory protocols in protocols.io, where a protocol can be assigned its own identifier (DOI) such that it can be cited independently in the future. For instructions see: http://journals.plos.org/plosone/s/submission-guidelines#loc-laboratory-protocols

We look forward to receiving your revised manuscript.

Kind regards,

Joël Mossong

Academic Editor

PLOS ONE

Journal Requirements:

2. Thank you for providing the following Funding Statement:"Funding: Sanofi Pasteur."

We note that one or more of the authors are employed by a commercial company: name of commercial company.

a) Please provide an amended Funding Statement declaring this commercial affiliation, as well as a statement regarding the Role of Funders in your study. If the funding organization did not play a role in the study design, data collection and analysis, decision to publish, or preparation of the manuscript and only provided financial support in the form of authors' salaries and/or research materials, please review your statements relating to the author contributions, and ensure you have specifically and accurately indicated the role(s) that these authors had in your study. You can update author roles in the Author Contributions section of the online submission for

b). Please also provide an updated Competing Interests Statement declaring this commercial affiliation along with any other relevant declarations relating to employment, consultancy, patents, products in development, or marketed products, etc. 

Additional Editor Comments (if provided):

Reviewers' comments:

Reviewer's Responses to Questions

**Comments to the Author**

1. Is the manuscript technically sound, and do the data support the conclusions?

Reviewer #1: Yes

Reviewer #2: Yes

Reviewer #3: No

2. Has the statistical analysis been performed appropriately and rigorously? 

Reviewer #1: Yes

Reviewer #2: Yes

Reviewer #3: No

3. Have the authors made all data underlying the findings in their manuscript fully available?

Reviewer #1: No

Reviewer #2: No

Reviewer #3: No

4. Is the manuscript presented in an intelligible fashion and written in standard English?

Reviewer #1: Yes

Reviewer #2: Yes

Reviewer #3: Yes

5. Review Comments to the Author

Reviewer #1: The authors estimated influenza-associated mortality in the Philippines, which added one more component to understand the global burden of influenza. In general, this paper was well written, and I have some comments below.

1. What contributes to the sharp peak in death in 2008 for children aged 5-9 years?

2. The authors could consider to add a dummy variable in the regression model to adjust for the impact of Typhoon Haiyan, like what has been done to adjust for SARS outbreak.

3. Why would the effects of temperature and relative humidity be opposite in different age groups?

4. The mathematical formula is incorrect. In the negative binomial model, there should be a log link between dependent and independent variables.

5. In table 1, why there is a substantial decrease of samples tested in 2014 and 2015. Is there any change of influenza surveillance system in Philippines?

6. The axes of the figures could be revised to make the figures more readable.

Reviewer #2: This manuscript estimates Influenza-associated excess mortality in the Philippines during the period 2006 through 2015 using negative binomial regression models. As the authors indicate, mortality burden of influenza in the Philippines has not been quantified before thus findings from this manuscript will help to inform public health policies and strategies for the control of influenza.

Here are some comments that the authors could consider to help the readers to better understand their study and the findings that they present.

1. In line 78: The authors state, “No adjustment was made for under-registration of deaths”. Before that, the authors indicate that deaths “must be registered within 48 hours”. Isn’t this a contradiction? It would be of help to the readers if the authors could state why no adjustments were made for under-registration. Is it that no data are available to quantify deaths that are not registered, or perhaps that there is so much variation by site/hospital or region in the country to allow for meaningful adjustments?

2. The authors indicate that GISRS data were collected through ILI and SARI cases at sentinel sites. Can they include the case definitions used for ILI and SARI or provide appropriate references for the readers?

3. I note that the authors used a weekly time-series to model the overall and age-specific EMRs associated with influenza. Did the authors consider using time-lagged independent variables (particularly for meteorological variables and influenza activity)? This is not stated in the text and one would expect a time-lagged effect of these variables on mortality.

4. Also, is there a particular reason why the meteorological variables (rainfall, temperature, and humidity) were included in the model to estimate EMRS? Has there been data to suggest that these are important for the Philippines? Did inclusion of these variables result in better fitting models, across the age groups assessed?

5. In lines 118-119, the authors state the “Other missing data were not replaced in the principal analyses”. This is somehow ambiguous. Could the authors state what these data are?

6. Reading through the methods, I assume that the data from the 2009 pandemic period were included in the analyses, is this correct? If so, I would suggest that the authors rerun the analyses with the pandemic data excluded. This is particularly important if they are seeking to estimate the mean annual EMRs associated with influenza. A quick look at the data suggests an additional few hundred deaths (~300 deaths) annually if the data from the pandemic period are included.

7. Related to that, I suggest that the authors only model the estimates using the imputed deaths for week 45 in 2013 (when the typhoon Haiyan occurred). Regardless of the fact that they conducted sensitivity analyses, it is clear that the spike in deaths during that week was out of the ordinary and thus it would only make sense if it were excluded.

8. The authors mention that the 95% CIs were estimated using bootstrapping methods with 1,500 iterations. However, I note that the CIs are very narrow, particularly for data among young children <5 years. Could the authors look again at this, and perhaps comment about it in the discussion?

9. In lines 208-209, the authors state “…. and influenza was the cause of approximately one in every 100 deaths”. This is not explicit in the results section but I suppose that this is based on dividing the total mean annual all-cause deaths by the mean annual influenza-associated deaths. Could the authors try to make this more explicit in the text? This further highlights why you should not include the 2009 pandemic data in your EMRs models.

Reviewer #3: The authors intend to quantify mortality associated to influenza in the Philippines, which is very important for public health and prevention. Hence, this is a very important paper.

Some general comments:

The authors use three metrological measures, rain, temperature and humidity, which each also have seasonal variation. Therefore, the inclusion of the yearly and half-yearly sines must have been included to adjust for residual seasonality not covered by these metrological variables.

Argue why a negative binomial regression (compensate for over-dispersion – what about under-dispersion?)

Additive or multiplicative model? – link function

Selection of elements to be included in the model is based on having positive A and B coefficients – argue why - and secondly the lowest AIC. However, in the S2 Table I miss this information.

Figure 2 show huge peaks in number of deaths associated to influenza in week 45 2013 (The typhoon Haiyan), This I don’t understand.

- Was there a huge peak in positive influenza samples that week? – I do not believe so, probably none or very few samples were sampled in that week.

- As there are no peaks in the model (red line) in figure 1, and the number of deaths associated to influenza was calculated as the prediction from the full model minus the prediction by the same model, but with A and B set to zero. There should not be calculated peaks.

This indicate that the calculation of influenza-associated number of deaths is wrong!

Would be nice to show both the full models and the models with A and/or B as zero in graph 1.

Suggest including graphs showing the A and B positive percentages over calendar time used in the model and as supplementary the metrological parameters.

The authors intend to compare all-cause influenza-associated mortality with cause-specific influenza-associated mortality. The most commonly used cause-specific mortality is respiratory (ICD10 …) e.g. references 1 and 2. The authors only look at J10 and J11 (influenza the main cause of deaths). This is of cause interesting, but it would have been of more interest, if they (also) had looked at respiratory cause of deaths, and made serious comparisons, for example if there is a general relation e.g. has it been suggested that influenza-associated mortality estimated using all-cause is the double of respiratory.

Alternatively, the authors could leave out cause-specific influenza-associated mortality, and write another article comparing all-cause and cause-specific.

Suggest to include a typhoon Haiyan parameter in the model: 1 in week 45 2013, else 0.

The Philippines consist of many islands with varying population. The metrological stations are properly distributed more-or-less evenly over the whole area, why average metrological measures should be population weighted. Likewise for the influenza data.

Is this possible? – if not, this should be discussed as a limitation.

Minor comments:

Page 3, line 64: I believe ‘mortality has not been accurately quantified’ attribute too much faith in the model. Suggest to leave out ‘accurately’.

Page 4, line 66-67: ‘… and compare …’. Suggest ‘… and compare all-cause estimates with influence cause-specific estimates’.

Page 4, line 78: Is it correct that all deaths in the Philippines are registered with cause of deaths within 48 hours? – Faster than in any other country, I know of.

Page 4, line 81-86: What do you mean by and what is the difference between ILI sentinel sites and SARI (Severe acute respiratory infections) sites? – how many of each?

Page 5, line 93. I cannot find ‘average weekly temperature’ in reference 13, only average daily min and max temperatures.

Page 5, line 105: Ok to use a polynomial spline, but why 6?

Page 6, line 118: ‘Other missing data …’ - how many?

Page 13, line 212-215. I believe it is highly surprising that this studies all-cause influenza-associated mortality estimates are consistent with the respiratory-cause-specific estimates from Iuliano et al. study

Page 13, line 223: I would not use the word ‘accurately’, but something like our model fitted data well

Page 13, line 223-4: Sometimes ‘official deaths registry statistics’ stand for cause-specific and here for all-cause. Suggest using all-cause and cause-specific.

Page 13, line 224: It is not correct that your model predicted the typhoon Haiyan peak! – see figure 1

Page 14-15, line 252-255: You might have used, what is often called the Goldstein index: ILI-rate * positive-percentage, where the ILI rate reflect the population dynamics and the positive-percentage limit the ILI-rate to influenza i.e. exclude other circulating respiratory pathogens.

6. PLOS authors have the option to publish the peer review history of their article (what does this mean?). If published, this will include your full peer review and any attached files.

Reviewer #1: Yes: Xiling Wang

Reviewer #2: No

Reviewer #3: No

---

## [Author Response · Author response to Decision Letter 0]

9 May 2020

Point-by-point response to Reviewers’ comments

Reviewer #1: 

The authors estimated influenza-associated mortality in the Philippines, which added one more component to understand the global burden of influenza. In general, this paper was well written, and I have some comments below.

1. What contributes to the sharp peak in death in 2008 for children aged 5-9 years?

Upon reviewing our files, we found out that this sharp peak was due to a typographical error that arose when we transferred our estimates to Excel to generate the graphs. Apologies for this inconvenience and thank you for noticing. Please rest assured that all estimates were done appropriately – it was just in the graphing that the error arose.

2. The authors could consider to add a dummy variable in the regression model to adjust for the impact of Typhoon Haiyan, like what has been done to adjust for SARS outbreak.

Thank you very much for your suggestion. We revised the model to include a dummy for Haiyan and a dummy for the pandemic year. Our new modelling algorithm is documented in S1 Text and S2 Table.

3. Why would the effects of temperature and relative humidity be opposite in different age groups?

While these effects are interesting and could be investigated, we feel that this could be explored and further discussed in a different study. We are primarily interested in flu activity as predictors of deaths and these meteorologic controls were included as previous studies. We feel that the changes in direction are secondary to the goal of improving model fit and we have shown that the fits of our models that include these parameters fit the data well. 

4. The mathematical formula is incorrect. In the negative binomial model, there should be a log link between dependent and independent variables.

Thank you for letting us know. We rewrote it as: 

E[Y_t ]=exp{β_0+β_1 t+β_2 t^2+β_3 t^3+β_4 t^4+β_5 t^5+β_6 t^6+ β_7 [Influenza A]_t+β_8 [Influenza B]_t + β_9 [Rainfall]_t+β_10 [Mean Temperature]_t+β_11 [Relative Humidity]_t+β_12 [Haiyan]_t+β_13 [Pandemic]_t+β_14 [sin⁡(2πt/52) ]+β_15 [cos⁡(2πt/52) ]+β_16 [sin⁡(2πt/26) ]+β_17 [cos⁡(2πt/26) ]+e_t }

where e is the error term that follows exp(e_t )~Gamma(1⁄α,α), and α is the overdispersion parameter following past research. 

5. In table 1, why there is a substantial decrease of samples tested in 2014 and 2015. Is there any change of influenza surveillance system in Philippines?

According to the Research Institute of Tropical Medicine, the National Influenza Center, the data collection was scaled down from 2012 onward. However, this should not be an issue since we used positivity rates instead of raw counts.

Source: http://ritm.gov.ph/reference-laboratories/national-reference-laboratories/influenza-and-other-respiratory-viruses/

6. The axes of the figures could be revised to make the figures more readable.

We have done so.

Reviewer #2: 

This manuscript estimates Influenza-associated excess mortality in the Philippines during the period 2006 through 2015 using negative binomial regression models. As the authors indicate, mortality burden of influenza in the Philippines has not been quantified before thus findings from this manuscript will help to inform public health policies and strategies for the control of influenza.

Here are some comments that the authors could consider to help the readers to better understand their study and the findings that they present.

1. In line 78: The authors state, “No adjustment was made for under-registration of deaths”. Before that, the authors indicate that deaths “must be registered within 48 hours”. Isn’t this a contradiction? It would be of help to the readers if the authors could state why no adjustments were made for under-registration. Is it that no data are available to quantify deaths that are not registered, or perhaps that there is so much variation by site/hospital or region in the country to allow for meaningful adjustments?

Apologies for the confusion. We meant that the policy dictates that deaths should be recorded within 48 hours. But this does not always happen. In fact, there is one small study that quantifies the extent of underreporting. This was done in only one province in the Philippines so we cannot confidently say that the extent is true nationwide. Given the uncertainty of the degree of underreporting according to location and according to cause of disease, we have decided to not adjust for underreporting. Our findings would likely underestimate the burden of disease due to flu. We have included these points in the limitations section of our discussion. Moreover, since the analysis does not depend on 48 hour policy, we decided to remove that information to avoid confusing the readers. 

Source: Carter, K. L., Williams, G., Tallo, V., Sanvictores, D., Madera, H., & Riley, I. (2011). Capture-recapture analysis of all-cause mortality data in Bohol, Philippines. Population health metrics, 9(1), 9.

2. The authors indicate that GISRS data were collected through ILI and SARI cases at sentinel sites. Can they include the case definitions used for ILI and SARI or provide appropriate references for the readers?

We have done as you suggested in lines 88-94: “In the surveillance, ILI was defined as an acute respiratory infection with measured fever of ≥38C° and cough with onset within the last 10 days while severe acute respiratory infection (SARI) was defined as an acute respiratory infection with history of fever or measured fever of ≥38C° and cough with onset within the last 10 days and requires hospitalization.” Our definitions are aligned with the definitions by FluNet and WHO. 

3. I note that the authors used a weekly time-series to model the overall and age-specific EMRs associated with influenza. Did the authors consider using time-lagged independent variables (particularly for meteorological variables and influenza activity)? This is not stated in the text and one would expect a time-lagged effect of these variables on mortality.

Thank you for your suggestion. We tried introducing up to two-period lags for the flu variables and the model with two lags yielded the best model fit. We did not introduce lags for the meteorological controls since this is not commonly done in similar studies (e.g. Aungkalanon et al 2015; Chow et al 2006). 

Source: Aungkulanon, S., Cheng, P. Y., Kusreesakul, K., Bundhamcharoen, K., Chittaganpitch, M., Margaret, M., & Olsen, S. (2015). Influenza‐associated mortality in T hailand, 2 006–2011. Influenza and other respiratory viruses, 9(6), 298-304.

Chow, A., Ma, S., Ling, A. E., & Chew, S. K. (2006). Influenza-associated deaths in tropical Singapore. Emerging infectious diseases, 12(1), 114.

4. Also, is there a particular reason why the meteorological variables (rainfall, temperature, and humidity) were included in the model to estimate EMRS? Has there been data to suggest that these are important for the Philippines? Did inclusion of these variables result in better fitting models, across the age groups assessed?

Many previous studies (e.g. Aungkulanon et al 2015; Chow et al 2006) included meteorological controls in their models. A local paper by Lucero et al (2016) points out that flu circulates year-round, but is more pronounced from June to November. Given that the extant literature suggests that meteorological controls are important in modelling EMR, we included meteorological controls in all our models by default. 

Source: Aungkulanon, S., Cheng, P. Y., Kusreesakul, K., Bundhamcharoen, K., Chittaganpitch, M., Margaret, M., & Olsen, S. (2015). Influenza‐associated mortality in T hailand, 2 006–2011. Influenza and other respiratory viruses, 9(6), 298-304.

Chow, A., Ma, S., Ling, A. E., & Chew, S. K. (2006). Influenza-associated deaths in tropical Singapore. Emerging infectious diseases, 12(1), 114.

Lucero, M. G., Inobaya, M. T., Nillos, L. T., Tan, A. G., Arguelles, V. L. F., Dureza, C. J. C., ... & Rodriguez, T. (2016). National Influenza Surveillance in the Philippines from 2006 to 2012: seasonality and circulating strains. BMC infectious diseases, 16(1), 762.

5. In lines 118-119, the authors state the “Other missing data were not replaced in the principal analyses”. This is somehow ambiguous. Could the authors state what these data are?

Apologies for the confusion. There is no other missing data apart from the positivity rate of 2008 week 16. Therefore, we deleted this sentence.

6. Reading through the methods, I assume that the data from the 2009 pandemic period were included in the analyses, is this correct? If so, I would suggest that the authors rerun the analyses with the pandemic data excluded. This is particularly important if they are seeking to estimate the mean annual EMRs associated with influenza. A quick look at the data suggests an additional few hundred deaths (~300 deaths) annually if the data from the pandemic period are included.

Thank you for this excellent point. We tried running the models without the pandemic year but the results seem to be close to the original EMR of 5.33 per 100,000. In fact, we got a slightly higher EMR of 5.81 when we excluded the pandemic year. Thus, we decided to just keep the whole dataset and run the models with the pandemic year controlled for. After controlling for the pandemic, our new runs yielded a more conservative estimate of EMR of 5.03.

7. Related to that, I suggest that the authors only model the estimates using the imputed deaths for week 45 in 2013 (when the typhoon Haiyan occurred). Regardless of the fact that they conducted sensitivity analyses, it is clear that the spike in deaths during that week was out of the ordinary and thus it would only make sense if it were excluded.

Thank you for pointing this out. We agree that Haiyan caused an unusual spike in death for week 45 of 2013. However, we felt that it would be better to keep the data as it is for the base case, and then run the modelling on imputed values. We dealt with it by adding a dummy variable for Haiyan as a control variable instead. We conducted a sensitivity analyses that either excludes this week or used an imputed value instead and found that the results did not change to a large degree.

8. The authors mention that the 95% CIs were estimated using bootstrapping methods with 1,500 iterations. However, I note that the CIs are very narrow, particularly for data among young children <5 years. Could the authors look again at this, and perhaps comment about it in the discussion?

Save for the number of iterations, we have little control over the resulting CIs since bootstrapping is entirely data driven. We have double-checked our code and re-ran the analysis to ensure correctness of analysis procedure. <up to Kent if they want to discuss but mention it here if added in discussion> 

9. In lines 208-209, the authors state “…. and influenza was the cause of approximately one in every 100 deaths”. This is not explicit in the results section but I suppose that this is based on dividing the total mean annual all-cause deaths by the mean annual influenza-associated deaths. Could the authors try to make this more explicit in the text? This further highlights why you should not include the 2009 pandemic data in your EMRs models.

This was calculated by dividing the average excess mortality per year by the annual average all cause mortality. We made it more explicit on lines 185-186: “Influenza was estimated to account for a mean of 5,347 excess deaths per year (about 1.1% of the average annual all-cause deaths for the study period)…”

Reviewer #3: 

The authors intend to quantify mortality associated to influenza in the Philippines, which is very important for public health and prevention. Hence, this is a very important paper.

Some general comments:

1. The authors use three metrological measures, rain, temperature and humidity, which each also have seasonal variation. Therefore, the inclusion of the yearly and half-yearly sines must have been included to adjust for residual seasonality not covered by these metrological variables.

We agree. Therefore, we added the sine and cosine terms. 

2. Argue why a negative binomial regression (compensate for over-dispersion – what about under-dispersion?) Additive or multiplicative model? – link function

We also considered using Poisson models in the analysis but results from using negative binomial models showed evidence of over-dispersion (alpha is not equal to zero and the point estimate is greater than 1). We have added these details in the methods.

Line 112: “Negative binomial regression was used instead of the Poisson regression since deaths were over-dispersed.”

As is usual practice, we used a log-link in the models. We apologize for failing to include this detail. We have updated the formula to be more explicit about the link function used.

3. Selection of elements to be included in the model is based on having positive A and B coefficients – argue why - and secondly the lowest AIC. However, in the S2 Table I miss this information.

We agree with this comment and have decided to just use the model fit (AIC) as the sole criterion to select the model. We have repeated the model selection and have presented updated results.

In addition, based on the comments of the other reviewers, we decided to introduce up to two lags of the flu variables and control for the Haiyan week and the 2009 pandemic. To facilitate readability of the paper, we opted to show just the results for the selected model.

4. Figure 2 show huge peaks in number of deaths associated to influenza in week 45 2013 (The typhoon Haiyan), This I don’t understand. - Was there a huge peak in positive influenza samples that week? – I do not believe so, probably none or very few samples were sampled in that week.

We agree that this appears as a surprising result. We feel that this has been addressed by the results of the sensitivity analysis and have been mentioned in the Discussion:

Results, Lines 209-211: 

“A peak in influenza deaths also occurred in the week of Typhoon Haiyan (week 45 of 2013, figure not shown). However, in a sensitivity analysis, the EMRs were not considerably changed by using an imputed value for all-cause deaths at week 45 of 2013 (S4 Table).”

Discussion, Lines 250-260: 

“Our regression models also projected a peak in influenza deaths in the week of typhoon Haiyan. Although this peak could be related to struggling health services and increased transmission of infections following the typhoon [27], it might also be partly influenced by the increased all-cause deaths at this time. By nature of regression analyses, trends of both dependent and independent variables are summarized by a fixed coefficient. Therefore, the peak in all-cause deaths – the dependent variable – could have translated to upward trends in estimated influenza deaths, even if the influenza positivity rates of tested samples were not elevated during the same period. Nonetheless, the overall EMRs were largely unaffected by this exceptional event since we controlled for the Typhoon Haiyan week in the main analyses and in the sensitivity analyses, where we used an imputed value to replace the peak in all-cause deaths for the Haiyan week.”

5. As there are no peaks in the model (red line) in figure 1, and the number of deaths associated to influenza was calculated as the prediction from the full model minus the prediction by the same model, but with A and B set to zero. There should not be calculated peaks.

This indicate that the calculation of influenza-associated number of deaths is wrong!

While there was a peak in actual all-cause mortality in 2014 due to Haiyan, we do not think that models are made to predict all the peaks and troughs as exactly as the real data. We would run the risk of overfitting the model if we try to predict every single peak and trough. However, since Haiyan is a special event that needs to be accounted for, we have revised the modelling strategy to control for the Haiyan week. We found that our updated results did not lead to drastically different estimates. We also found that the original models without the Haiyan and pandemic controls and the revised models perform well as its predicted all-cause mortality is 90+% correlated with the actual all-cause mortality. For the final paper, we’ve decided to include the Haiyan variable in the main model.

6. Would be nice to show both the full models and the models with A and/or B as zero in graph 1.

We feel that adding this detail in the main text might be too much but we have included these graphs as part of the supplemental material.

7. Suggest including graphs showing the A and B positive percentages over calendar time used in the model and as supplementary the metrological parameters.

We have done so as a supplementary file. See S1 Figure.

8. The authors intend to compare all-cause influenza-associated mortality with cause-specific influenza-associated mortality. The most commonly used cause-specific mortality is respiratory (ICD10 …) e.g. references 1 and 2. The authors only look at J10 and J11 (influenza the main cause of deaths). This is of cause interesting, but it would have been of more interest, if they (also) had looked at respiratory cause of deaths, and made serious comparisons, for example if there is a general relation e.g. has it been suggested that influenza-associated mortality estimated using all-cause is the double of respiratory.

Alternatively, the authors could leave out cause-specific influenza-associated mortality, and write another article comparing all-cause and cause-specific.

We agree with this point since other papers have used cause-specific causes of death in the modelling. There is however the problem of inaccurate coding of causes of death. Many doctors were not cognizant to the right way of coding cause of death on the death certificates (Hufana et al 2009). Due to this uncertainty in coding, we went with the conservative route of just using all-cause mortality instead of the cause-specific deaths because there is more certainty in the counts of total deaths than cause-specific deaths.

Source: Carter, K.L., Williams, G., Tallo, V. et al. Capture-recapture analysis of all-cause mortality data in Bohol, Philippines. Popul Health Metrics 9, 9 (2011) doi:10.1186/1478-7954-9-9

Hufana L, Cajita J, Morante L, Lopez J, Tan CL, Mikkelsen L, et al. Assessing the production, quality and use of national vital statistics: a case study of the Philippines. 2009. [cited 7 Mar 2019]. Available from: https://crvsgateway.info/file/7572/98

9. Suggest to include a typhoon Haiyan parameter in the model: 1 in week 45 2013, else 0.

We agree with this suggestion and have revised our model.

10. The Philippines consist of many islands with varying population. The metrological stations are properly distributed more-or-less evenly over the whole area, why average metrological measures should be population weighted. Likewise for the influenza data.

Is this possible? – if not, this should be discussed as a limitation.

We had to drop weather stations that had 6-months’ worth of missing data so in effect, our meteorological averages were computed from 37 instead of 52 weather stations. We therefore cannot conclusively say that the weather stations we ended up using are proportionate to the population. Furthermore, using a weighted mean calls for another layer of assumption, specifically, the weight to be used. Because of these reasons, we decided to use the simple mean instead. 

Similarly, the FluNet does not provide the information on a sentinel-site basis. In addition, there is no indication that the sentinel sites are distributed evenly throughout the country. Lucero et al (2016) states that the sites were only present in 13 out of 17 regions in the country. Thus, we cannot weigh the flu positivity rates by population size. 

Following the previous suggestions, we added the following sentences to specify that we had limitations with the weather station and flu surveillance data.

Lines 283-290: “Third, FluNet data may not be nationally-representative. Sentinel surveillance sites were only present in 13 out of 17 regions in the country [8] and there is no indication that sites were distributed evenly across the country. Despite its limitations, FluNet data is the only source of influenza activity data of its scope in the country to date. Fourth, because of missingness of the data, we had to drop 15 out of 52 weather stations for the meteorological variables. We believe that this is not an issue since the meteorological variables were merely controls, not the independent variable of interest.” 

Lucero, M.G., Inobaya, M.T., Nillos, L.T. et al. National Influenza Surveillance in the Philippines from 2006 to 2012: seasonality and circulating strains. BMC Infect Dis 16, 762 (2016) doi:10.1186/s12879-016-2087-9

Minor comments:

11. Page 3, line 64: I believe ‘mortality has not been accurately quantified’ attribute too much faith in the model. Suggest to leave out ‘accurately’.

We deleted the word “accurately.”

12. Page 4, line 66-67: ‘… and compare …’. Suggest ‘… and compare all-cause estimates with influence cause-specific estimates’.

Apologies for being unclear. What we meant to say was that we wanted to compare the flu mortality estimates with the official flu mortality data from the death registry data. So that it will be clearer, we revised the sentence into (lines 66-69): “Here, we used negative binomial regression models to estimate the influenza-attributable mortality in the Philippines from 2006 to 2015, and compared the flu mortality estimates with death registry data to understand under-reporting of influenza deaths.” 

13. Page 4, line 78: Is it correct that all deaths in the Philippines are registered with cause of deaths within 48 hours? – Faster than in any other country, I know of.

Death registration in the Philippines is not always implemented as intended by policy. A study on death registration in one province showed that only 77% of deaths are recorded. However, since we do not know much about underreporting of deaths, we chose not to adjust it in our current modelling exercise.

Source: Carter, K.L., Williams, G., Tallo, V. et al. Capture-recapture analysis of all-cause mortality data in Bohol, Philippines. Popul Health Metrics 9, 9 (2011) doi:10.1186/1478-7954-9-9

14. Page 4, line 81-86: What do you mean by and what is the difference between ILI sentinel sites and SARI (Severe acute respiratory infections) sites? – how many of each?

ILI sentinel surveillance sites are health centers and hospital outpatient departments while SARI sentinel surveillance sites are hospital inpatient departments. The FluNet data did not specify how many sites there were, but according to this study which documented flu surveillance in the Philippines, from 2006 to October 2008 there were 18 health centers and 18 outpatient departments. Then by 2009, 16 more health centers were added. As of 2009, there were 52 sentinel sites. Unfortunately, we cannot find a dossier that gives an update on the number of sentinel sites from 2009-2015, 2015 being the last data point of our study. We have added these details in the methodology.

Lucero, M.G., Inobaya, M.T., Nillos, L.T. et al. National Influenza Surveillance in the Philippines from 2006 to 2012: seasonality and circulating strains. BMC Infect Dis 16, 762 (2016) doi:10.1186/s12879-016-2087-9

15. Page 5, line 93. I cannot find ‘average weekly temperature’ in reference 13, only average daily min and max temperatures.

We apologize for the confusion. The World Meteorological Organization has no specific guideline on average weekly temperature. We just applied their recommendation for average daily temperature to come up with the weekly temperature. Specifically, on page 78 of their manual, they say that “All ordinary climatological stations observe a daily maximum and minimum temperature … Hence, the recommended methodology for calculating average daily temperature is to take the mean of the daily maximum and minimum temperatures.”

To make our point clearer, we revised the concerned sentence into (lines 104-107): “The average weekly temperature was calculated by taking the mean of the weekly maximum and weekly minimum, following the logic of the World Meteorological Organization’s recommended method of computing for average daily temperature.”

Source: World Meteorological Organization. Guide to Climatological Practices 2011. [cited 16 July 2019]. Available from: http://www.wmo.int/pages/prog/wcp/ccl/guide/documents/WMO_100_en.pdf

16. Page 5, line 105: Ok to use a polynomial spline, but why 6?

We only have 6 years’ worth of data so we thought of keeping our polynomials up to 6. We also added the polynomial terms one-by-one (up to 6 polynomial terms) until we reach the model that provides the best fit.

17. Page 6, line 118: ‘Other missing data …’ - how many?

We did not have other missing data, apart from the number of tested samples for week 16 of 2008. We therefore deleted the “Other missing data…” sentence.

18. Page 13, line 212-215. I believe it is highly surprising that this studies all-cause influenza-associated mortality estimates are consistent with the respiratory-cause-specific estimates from Iuliano et al. study

We disagree that our results are surprising in this aspect. While we modelled all-cause mortality, we expect that most of these deaths would be respiratory deaths and cardiovascular deaths since these are the top causes of mortality in our population and other studies have linked flu to respiratory and cardiovascular deaths. Since our outcome includes respiratory deaths, our estimates should be close to Iuliano et al’s estimates.

Source: http://www.healthdata.org/sites/default/files/files/country_profiles/GBD/ihme_gbd_country_report_philippines.pdf

19. Page 13, line 223: I would not use the word ‘accurately’, but something like our model fitted data well

We removed “accurately” from our discussions; see line 247.

20. Page 13, line 223-4: Sometimes ‘official deaths registry statistics’ stand for cause-specific and here for all-cause. Suggest using all-cause and cause-specific.

We agree with the suggestion and have revised the sentence to use “all-cause” mortality rather than ‘official deaths registry statistics’.

21. Page 13, line 224: It is not correct that your model predicted the typhoon Haiyan peak! – see figure 1

We apologize for this reporting error. We have deleted that sentence in the manuscript. 

22. Page 14-15, line 252-255: You might have used, what is often called the Goldstein index: ILI-rate * positive-percentage, where the ILI rate reflect the population dynamics and the positive-percentage limit the ILI-rate to influenza i.e. exclude other circulating respiratory pathogens.

To clarify, we did not use the Goldstein Index. What we used was the flu positivity rate which was computed by (specimens tested positive)/(total specimens tested). We do not have good estimates of daily ILI rates in the population precluding use of the Goldstein index.

---

## [Decision Letter · Decision Letter 1]

2 Jun 2020

Influenza-associated excess mortality in the Philippines, 2006-2015

PONE-D-19-24478R1

Dear Dr. Cheng,

We are pleased to inform you that your manuscript has been judged scientifically suitable for publication and will be formally accepted for publication once it complies with all outstanding technical requirements.

With kind regards,

Joël Mossong

Academic Editor

PLOS ONE

Additional Editor Comments (optional):

Reviewers' comments:

Reviewer's Responses to Questions

**Comments to the Author**

1. If the authors have adequately addressed your comments raised in a previous round of review and you feel that this manuscript is now acceptable for publication, you may indicate that here to bypass the “Comments to the Author” section, enter your conflict of interest statement in the “Confidential to Editor” section, and submit your "Accept" recommendation.

Reviewer #1: All comments have been addressed

Reviewer #2: All comments have been addressed

Reviewer #3: All comments have been addressed

2. Is the manuscript technically sound, and do the data support the conclusions?

Reviewer #1: Yes

Reviewer #2: Yes

Reviewer #3: Yes

3. Has the statistical analysis been performed appropriately and rigorously? 

Reviewer #1: Yes

Reviewer #2: Yes

Reviewer #3: Yes

4. Have the authors made all data underlying the findings in their manuscript fully available?

Reviewer #1: Yes

Reviewer #2: (No Response)

Reviewer #3: Yes

5. Is the manuscript presented in an intelligible fashion and written in standard English?

Reviewer #1: Yes

Reviewer #2: Yes

Reviewer #3: Yes

6. Review Comments to the Author

Reviewer #1: (No Response)

Reviewer #2: (No Response)

Reviewer #3: No further comments, the manuscript have been correctly and thoroughly revised and I believe it's ready for publication.

7. PLOS authors have the option to publish the peer review history of their article (what does this mean?). If published, this will include your full peer review and any attached files.

Reviewer #1: No

Reviewer #2: No

Reviewer #3: No

---

## [Editor Report · Acceptance letter]

4 Jun 2020

PONE-D-19-24478R1 

Influenza-associated excess mortality in the Philippines, 2006-2015 

Dear Dr. Cheng:

I'm pleased to inform you that your manuscript has been deemed suitable for publication in PLOS ONE. Congratulations! Your manuscript is now with our production department. 

Kind regards, 

on behalf of

Dr. Joël Mossong 

Academic Editor

PLOS ONE